# The Regulated Secretion and Models of Intracellular Transport: The Goblet Cell as an Example

**DOI:** 10.3390/ijms24119560

**Published:** 2023-05-31

**Authors:** Alexander A. Mironov, Galina V. Beznoussenko

**Affiliations:** Department of Cell Biology, IFOM ETS—The AIRC Institute of Molecular Oncology, Via Adamello, 16, 20139 Milan, Italy; galina.beznusenko@ifom.eu

**Keywords:** Golgi complex, intracellular transport, COPI, regulated secretion, secretory granule, mucin, mucus, ER-Golgi transport

## Abstract

Transport models are extremely important to map thousands of proteins and their interactions inside a cell. The transport pathways of luminal and at least initially soluble secretory proteins synthesized in the endoplasmic reticulum can be divided into two groups: the so-called constitutive secretory pathway and regulated secretion (RS) pathway, in which the RS proteins pass through the Golgi complex and are accumulated into storage/secretion granules (SGs). Their contents are released when stimuli trigger the fusion of SGs with the plasma membrane (PM). In specialized exocrine, endocrine, and nerve cells, the RS proteins pass through the baso-lateral plasmalemma. In polarized cells, the RS proteins secrete through the apical PM. This exocytosis of the RS proteins increases in response to external stimuli. Here, we analyze RS in goblet cells to try to understand the transport model that can be used for the explanation of the literature data related to the intracellular transport of their mucins.

## 1. Introduction

Regulatory secretion (RS) occurs under the influence of external stimuli, whereas constitutive secretion does not depend on external stimuli. RS is not found only in mammalian cells, but also in some protists; they are not found in plants and yeast [1,2]. RS is realized through the apical plasma membrane (APM) and baso-lateral PM (BLPM). RS based on the transfer of RS proteins through the APM exists in the mucous glands of different tracts, for instance, in digestive and respiratory tracts, namely, in acinar cells in pancreas, mammary, sweat, salivary and lacrimal glands, as well as in goblet cells. RS proteins are secreted via the BLPM in endocrine and neuroendocrine cells. SGs containing procollagen move through the BLPM and do not depend on external stimulus. In turn, cells using RS via their APM can be divided into those using a wide or a rather narrow APM [2]. In the pancreas and salivary glands, the ratio between the surface areas of the APM and BLPM are significantly smaller than in goblet cells. In the mucous glands of the colon wall, epithelial cells are similar to singly arranged goblet-shaped cells. In the trachea, goblet-shaped cells have a prismatic shape and are taller than in the intestine. Epithelial cells in the mucous glands of the trachea and in the Brunner glands of the duodenum consist of cells similar to goblet cells [3].

The knowledge of how salivary, sweat, and lacrimal glands secrete RS proteins is important for clinicians because the mucus layer lubricates and protects epithelial cells from pathogenic and harmful agents [2,4]. The absence of mucus or any defect in the mucus layer allows a large number of bacteria to come into contact with epithelial cells, causing an excessive immune response in the host [5,6,7]. Both the lack and overproduction of mucus can be harmful to human health. The first case is manifested in lung diseases characterized by a hypersecretory mucus phenotype, such as cystic fibrosis. For example, goblet cell hyperplasia (an increase in the number of cells) is a sign of asthma, whereas goblet cell hypertrophy, not hyperplasia, occurs in cystic fibrosis [8]. The most unclear mechanisms of the RS-based intracellular transport is noted for goblet cells.

In general, the RS cycle includes: (1) the synthesis of RS proteins in ER and their intracellular transport at the Golgi complex (GC) with the formation of immature SGs there; (2) the initial maturation of these SGs; (3) the delivery of these SGs closer to the PM; (4) the final maturation of SGs; (5) fusion of mature SGs with the PM; 6) and the removal of SG membranes from the APM [9,10,11]. In most tissue cells with RS, isolated Golgi stacks are absent [12,13,14,15,16,17,18,19,20,21]. One exception is the GC in pancreatic β-cells where, however, RS is released through the BLPM [22].

Importantly, the deletion of various SG genes does not exclude the expression of other SG genes or SG biogenesis (although protein processing and/or their secretion may be affected [23]). Moreover, the isolation of mutants devoid of RS and SGs in multicellular animal organisms has never been reported, which suggests an important role of RS and indicates the vital function of RS and that many genes contribute to the structure and function of RS. In the current review, we study the literature and our residual data concerning the structure of the GC and the mechanisms of transport through it in goblet cells, where protein secretion is organized according to RS through a wide APM. Additionally, we use some data presented during the submission stage of two our papers [24,25] and pay further attention to the aspects of the cellular and tissue biology of goblet cells, since the molecular aspects are perfectly described in the excellent studies of Gustafsson et al. [26,27]. 

## 2. Features of Goblet Cells

Stem cells localized at the base of the intestinal crypts produce four main cell types, namely, Paneth, intestinal epithelial, enteroendocrine, and goblet cells [7,28]. During differentiation, goblet cells develop the ability to produce and accumulate a significant amount of mucus [5,7]. Goblet cells lengthen and decrease in volume as they move into the surface epithelium. The average cell volume decreases from 128.8 microm3 for goblet cells in the basal third of the crypt to 541.3 microm3 for goblet cells on the surface of intestinal villus [29]. During the migration of goblet cells along the middle half of the villi, mucus granules are completely renewed twice [30].

The base of the goblet cell is narrow because two or more dendritic cells are located around it, which reduce the contact area of the dendritic cell with the basement membrane (Figure 1A,C). Consequently, the contact area of cells with the basement membrane is smaller than that of absorbing intestinal cells. In goblet cells, SGs are concentrated near the APM. The cytoplasm and organelles are located around the SG zone and below the nucleus (Figure 1B,D–G). The nucleus contains dispersed chromatin, whereas after standard chemical fixation, it exhibits condensed chromatin.

Tight junctions between the goblet cells of the enterocytes with microvilli demonstrated rather similar structure [24]. In 23% of these junctions, the number of protein filaments was less than 4, and in 30% of the cases, the depth (or width) of these junctions was less than 200 nm. The defective cross-linking of these threads, freely ending abluminal and highly fragmented threads, are often evident. After lanthanum exposure to the pre-fixed loops of the ileum for 1 h, weak lanthanum permeability was detected in 42% of these tight junctions. Microperoxidase, cytochrome C, and horseradish peroxidase did not pass through these tight junctions [31]. SGs with mature mucin were concentrated near the APM. The cytoplasm and organelles were located around the SG zone and below the nucleus.

Nyström et al. [32] identified canonical and non-canonical goblet cells. Canonical goblet cells express high levels of Atoh1, Muc2, Fcgbp, and Clca1, whereas non-canonical goblet cells exhibit a more enterocyte-like expression profile with higher levels of expression of Hes1, Dmbt1, Muc17, and ion channels. This seems to be consistent with the previous studies [27]. On the other hand, canonical and non-canonical goblet cells can simply represent the stages of mucus synthesis and secretion. At the electron microscope level, the first type of goblet cells does not have holes in the APM with small microvilli with a diameter of 100 nm [33]. Two populations of goblet cells and the fact that goblet cells form interdigitated contacts and tight junctions with absorptive enterocytes were found in preparation for the publication of the article by Sesorova et al. [24].

In these cells, SGs have almost the same size and the same electron density. The second type of goblet cells shows signs of mucus secretion. Their APM is ruptured and exhibits holes due to its fusion with the membrane of the most apically located SGs. After this fusion and mucus secretion, the secretory granules increase in size and lose their electron density (Figure 1E). Scanning electron microscopy revealed several points of mucus extrusion in the free apical part of the goblet cells [27,34].

The GC of the goblet cells synthesize mucins. Mucins are divided into two main subfamilies: mucins of the cell surface, which attach to the cell membrane and provide a carbohydrate-rich coating, and secreted mucins, which give the mucous gel its viscous properties. Evolutionary studies suggest that mucins are ancient, with glycoproteins or mucin-like domains identified in viruses, parasites, and fungi. The accumulation of mucus is slow. They appear as being packed in the form of mucins of secretory granules, which account for about 75% of the cytoplasm volume [35,36,37]. The granules mature to form highly concentrated mucins, which eventually fuse with the plasma membrane and are secreted into the extracellular domain. Glycosylation accounts for up to 80% of the mass of mice. [7,37,38]. The mobility of GFP-tagged MUC5AC was significantly lower than that of the others [39].

## 3. Features of the Golgi Complex in Goblet Cells

The endoplasmic reticulum (ER) is an organelle in which muscles are synthesized and N-linked glycosylation occurs. The best fluorescent image of the GC goblet cell is presented by Gustafsson et al. [26] in their Figures 1D,E and 2E. There, the marker 1-43FX can partially serve as a GC marker. Interestingly, Figure 1H presented by Gustafsson et al. [26] shows that there is no endosomal marker EEA1 inside the cluster of secretory granules. There are no significant differences in the organization of GC in the goblet cells of the jejunum and colon (our unpublished observations). The GC is located in the upper part of the nucleus and has a cylindrical or cup-shaped shape. The “cup” of GC is filled with SGs, which are more abundant in the apical part of the cell. Mitochondria are rarely found inside the “Golgi cup”. 

In goblet cells, ER exit sites (ERESs) are very small in size (if they exist at all) and number. The number of round profiles that exist is low. In a study of more than 100 sections of goblet cells, we found only one structure similar in organization to ERESs. The ratio between the rim surface area and that of cisterna bodies is much smaller here than in cells grown in vitro. The connections between the ER and GC are of a varicose type. They can not only be found near the edges of the Golgi cisternae, but also near the central part of the cisternae. We did not find COPII-like coated buds on the ER. Connections between the ER and Golgi cisternae contain varicosities. These connections were observed within stack wells. There are connections between medial Golgi cisternae and smooth ER. The smooth ER is attached from the cis-side to the GC and can be connected to the proximal cisternae itself through varicosities-containing tubules. In Figure 2B presented by Gustafsson et al. [26], the resolution of the images is low, and these connections cannot be determined. Mucins are transported to the GC, where they are promoted by the glycosylation to the size of 2.5 million Daltons [7,38,40].

In goblet cells, the GC stack consists of the cis-most cisterna (CMC) and 6–7 medial cisternae. The CMC is located on the external side of the GC cup, close to the ER cisternae and the BLPM. The CMC consists of both tubes and sheets that are connected to each other by these tubes. These sheets have many pores with a diameter equal to 30 nm. The CMC often has large holes where tubes connect the parts of the CMC that have the usual type of perforations [12]. The Golgi stacks are very long and often bend. Usually, the GC in goblet-shaped cells consists of seven cisternae. Cisternae near the cis-pole of the GC are more osmiophilic than medial ones. Golgi stacks, which are rather long, form a single ribbon. Using a three-dimensional EM reconstruction based on FIBSEM, Gustafsson et al. [26] described the cup-like shape of the Golgi ribbon. This Golgi glass" has irregular holes in its walls and bottom. Golgi stack isolates from the Golgi ribbon were not observed (Figure 2, Figure 3 and Figure 4).

There are very rare COPI-coated buds on Golgi cisternae. After the examination of 51 goblet cells, we found only three COPI-coated buds on Golgi cisternae. One of them was situated on the flat area of Golgi cisternae, not on its rim. Golgi vesicles range in size from 40 to 80 nm. Clusters of COPI (or clathrin-dependent) vesicles with a diameter of 50–55 nm and vesicles with a diameter of 42–44 nm were also observed. These small vesicles are already described in enterocytes [24]. The clusters were found between the medial cisternae and between medial cisternae and SGs. Some vesicles were in close contact with the cisterna membrane. This could indicate that their final fusion was suppressed [41] and may have been caused by the fact that either SNAP29 was missing from the SG zone [42] or there was no enough calcium ions (Figure 2A–H).

The external surface of these long Golgi stacks contained wells that appeared as crater-like conus with a tip directed towards the trans-side of the Golgi stack. Here, cisternae form circular rather large pores. The diameter of the openings is high. In the second and third ones the diameter is smaller. The rims formed in this well are used for the formation of ER–Golgi connections. The lateral walls of the Golgi cup are rarely perforated. These large holes are surrounded by Golgi cisternae which have rims.

The ratio of the area of the edges and total area of the reservoirs sharply decreases compared to, for example, HeLa cells in vitro. There is a complex interweaving of cisternae; however, there are always cisternae with osmophilic contents on the outside of the glassy Golgi ribbon. There are no connections between apparently heterogenous cisternae passing in a direction perpendicular to the stack plane, as in the case we described previously [43]. Usually, the distensions of Golgi cisternae containing collagen in fibroblasts or VLDL in hepatocytes are located near the rims of the cisternae and separated from its main body with pores [44].

Medial Golgi cisternae can form a spiral. The formation of spiral shapes by Golgi cisternae is visible mostly within wells. In the GC of goblet cells, the ratios of the surface area of the rim membrane and other cisternae membranes were significantly lower than in all other cells in vitro. The thickness of the space between the membrane of SGs was less than that between cisternae per se and between SGs. A similar thinning of the intercisternal space was also observed within zones where cisterna distension filled with mucin made contact with the membrane of normal cisternae. This indicates that, even for membranes where mucin accumulates, GGE and nuclear sugar transporters are lost. SG membranes form close contacts forming a pentalamellar structure instead of a heptalamellar one [25]. SGs can also fuse [12]. Prior to mucus secretion, all SGs exhibited almost the same electron densities and sizes. In goblet cells exhibiting breaks of the APM, distal (situated near the APM) SGs usually lost their electron density. Long stacks with wells, the absence of ERES, and COPII-coated buds on the rough ER cisternae, the pure development of CMC, and the absence of TMC were discovered as a result of studying images presented in the literature and the analysis of images from the preparation and submission of an article by Briata et al. [25].

Starting from the 3–4th cisternae, distensions began to form. Initially, these distensions were localized near the cisternal rims. Then, the distensions filled with mucus were observed in the central parts of the Golgi cisternae. A typical trans-most cisterna was not observed. Some of these distensions formed close contacts with the membranes of cisternae or SGs. The membranes of the SGs adhered to each other and to the cisternal distensions. In the zone of adhesion, membranes were formed as three osmiophilic layers with two light layers between them. We did not find clathrin-coated buds on the membranes of SGs. 

In goblet cells, the molecular mechanisms responsible for the biogenesis of the GC and SGs were examined only occasionally, and there was no significant amount of suitable information. For instance, it was shown that the colocalization between Rab3D and TGN38 was low, which is localized inside the TGN. Despite this, the partial colocalization of Rab3D with βCOP and the Golgi cis-marker, GM-130, were observed. A high level of colocalization was observed between Rab3D and Griffonia simplicifolia agglutinin II lectins and soy agglutinin, markers of medial and cis-Golgi, respectively [45].

In goblet cells, the GC is capable of sulfation. The activity of TFR-aza was detected in internal Golgi cisternae [46]. Interestingly, two terminal glycosyltransferases, sialyltransferase and alpha-1,3 N-acetylgalactosaminyltransferase of blood group A, were found to exhibit different subcomputations in goblet-shaped GC and absorbent intestinal cells. As expected, due to their role in terminal glycosylation, two glycosyltransferases and their products, sialic acid residues and blood group A substances, were localized in the cisternae of the GC in goblet cells. However, in contrast, they were found throughout the GC stack of neighboring absorbing cells, with the exception of a fenestrated first cis-most cisterna [47].

Galactose-specific lectin Ricinus communis I labels Golgi cisternae in goblet cells. The reaction was weak in the medial cisternae, and the cis-side of the stacks was mostly devoid of labels [48]. Sialic acid enriched with Limax flavus lectin was found in the GC and SGs [49]. Ultrastructural staining of colon goblet cells with peanut lectin conjugate with horseradish peroxidase showed that Golgi cisternae were selectively stained in these areas. The contents of SGs or Golgi vesicles lacked affinity for this conjugate [50]. 

In other cells with RS, the concentration of RS proteins increased during intra-Golgi transport [51,52]. RS proteins were subjected to polymerization/condensation in the intraluminal conditions characteristic of young SGs, which include a slightly acidic pH and an increased concentration of divalent cations [53,54]. As immature SGs mature, their intraluminal pH becomes more acidic [55]; the concentration of Ca^2+^ in the cisternal and SG lumen increases [56]. This change in the ionic medium facilitates the condensation of RS proteins. 

The condensation of RS proteins has also been described in the transGolgi elements (“condensing vacuoles”) and immature SGs [57,58]. This is sometimes observed in medial Golgi cisternae and even in the ER. The high concentration of Ca^2+^ and pH < 6.5 promote protein condensation in SGs. The reason for the precipitation of mucins might be the formation of polysaccharide chains with sialic acids at their ends. These sialic acids can form hydrogenic bonds at low pH levels [2]. In the GC of the main epithelial cells of seminal vesicles, the first detectable aggregation of RS proteins was observed in medial cisternae [2,59]. Additionally, in rat parotid acinar cells, brefeldin A does not affect existing SGs, but blocks the biogenesis of new SGs. This indicates that role of COPI is extremely important for the processing of RS proteins. After washing out the BFA, the Golgi stacks regenerate, and then the formation of SGs resumes [60]. 

## 4. Role of Calcium

Acetylcholine causes mucus secretion, mainly from the small intestine and colon crypts [61]. Cholinergic agonists stimulate mucus secretion, mainly from goblet crypt cells in the small intestine, as well as in proximal and distal colons in mice, whereas histamine induces mucus secretion in a mouse colon, but not in the small intestine, and prostaglandin E2 induces mucus secretion in the mouse small intestine, but not in the colon [26,27]. 

It is known that mucus secretion is a biological process that depends on calcium. The ryanodine receptor-2 mediates the release of Ca^2+^ from the ER, which leads to an increase in the concentration of Ca^2+^ in the areas adjacent to the ER. Inositol receptor 1-,4-,5-triphosphate (IP3) also mediates the release of Ca^2+^ from the ER. A member of the Ca^2+^ family of the neuronal sensory protein KChIP3 (potassium protein interacting with voltage-dependent channels 3, also called DREAM and Calsenilin) is localized in a pool of secretory mucin granules. The binding of Ca^2+^ to KChIP3, followed by the dissociation of KChIP3 from mature secretory granules, ensures the fusion of RS with APM and the subsequent release of mucin into the intestinal lumen. The mechanism of mucus secretion associated with KChIP3 is not tissue-specific [36,62]. Ca^2+^ comes from the ER, immediately after the cargo leaves the ER [63]. Calcium destabilizes contacts between the SG membranes [39]. This influx of calcium leads to the fusion of the secretory granule membrane and the APM cascade. Calcium intake can cause the destabilization and fusion of such tightly fitting membranes, and without SNAREs.

The entrance of calcium ions into the cytosol leads to the fusion of the membranes of SGs and APM. Membranes of SGs tightly attached to each other are easily subjected to fusion [64]. Sometimes, deeply located droplets fuse with each other, and a large vacuole forms in the apical cytoplasm. Empty SGs are paired. However, there are no SG channels consisting of several empty SG channels (Figure 1). Ca^2+^ induces the fusion of COPI vesicles with the SG membrane, which increases their surface area and possibly creates traps. Then, the inner surface of the last secretory granule becomes an APM. After endocytosis, these membranes pass into autophagosomes and can even reach the Golgi cisternae and then be secreted through the PM into intercellular space [65].

Synaptotagmin (Ca^2+^- and phospholipid-id-binding protein) is involved in the establishment of the Ca^2+^-dependence of the fusion process [66]. Rab27A, Rab27B, and SYTL2 are specific for goblet cells. In mice, Vamp8 is important for mucus secretion [27,67]. The fact that SGs are attached to each other allows them to merge only by increasing the concentration of Ca^2+^. 

## 5. The Role of Mucins

Constant mucus secretion is a key factor determining the structure and function of the mucosal barrier. The mucosal barrier is not a static physical barrier, and it was found that the inner mucus layer in the distal colon of mice is renewed every 1–2 h [5,6,10]. It has been estimated that mucus spontaneously grows at a rate of approximately 240 microns/h in humans and 100 microns/h in mice [62,68,69]. Mucin release is accelerated when goblet cells are exposed to powerful secretagogues, and it is influenced by many different factors, including neuropeptides, cytokines, and lipids [7,10]. The exocytosis of mucin granules is a Ca^2+^-regulated process; therefore, mucus secretion can be divided into two modes (basic secretion and stimulated secretion), depending on the involvement of calcium influx [62].

When secreted, the tightly packed mucin expands more than 1000 times, forming large meshes. In the colon, the content of SGs is released at a very low rate [33]. To date, 21 human mucin genes have been identified [8]. Mucins associated with the cell membrane are designated as MUC1, MUC3A/B, MUC4, MUC12, MUC13, MUC15, MUC17, MUC20, and MUC21. Secreted mucins can be divided into gel-forming mucins (MUC2, MUC5AC, MUC5B, MUC6, and MUC19), which are necessary for the formation of a mucosal barrier on the surfaces of the mucous membrane, and non-gel-forming mucins (MUC7, MUC8, and MUC9) [7,70,71,72,73].

The gel-forming mucins secreted by goblet cells, submucosal gland cells, or serous cells form a highly hydrated mucous gel and contribute to the lubrication of the surface of epithelial cells. In the intestine, a mucus layer mainly consisting of MUC2 forms an inner adhesive layer and an outer loose layer. The inner adhesive layer is “sterile” and consists of MUC2 multimers, which are presumably tightly packed to provide protection from commensal flora [11,74].

The outer layer of mucus is exposed to proteases and bacteria, which allows it to become less dense. It is also important for maintaining homeostasis, since fecal material in the intestine creates mechanical stress (McGuckin et al., 2011 [75]). Once in the airway lumen, these mucins can cross-link non-covalently, forming a physical barrier that can be easily moved by cilia. Non-covalent and covalent-dependent interactions form a lateral network [35,75,76,77].

## 6. The Role of the Apical Endocytosis in Goblet Cells

There is an opinion that apical endocytosis occurs in goblet cells. However, clathrin-coated buds are not observed in the APM of goblet cells (Figure 1D,F). Endocytosis of the lumen material by goblet cells in the colon was described in the 1980s, when the absorption of cationic ferritin was shown using electron microscopy. Cationized ferritin added to the intestinal lumen was observed inside goblet cells [78]. Colloidal thorium particles detect acidic mucopolysaccharides on the reverse side of the GC in Golgi vacuoles adjacent to the reverse side and in the mucous droplets of goblet cells. Thorium was not associated with other cellular organelles, that is, with the coarse endoplasmic reticulum, mitochondria, or nuclei. In addition, thorium concentrations were absent in neighboring epithelial cells; although, the Golgi regions of these cells apparently contained slightly more thorium than could be explained by the background data [46]. It is important to note that there is no mechanism that blocks the ingress of intestinal contents into an empty secretory granule. MVBs exhibited a standard shape (Figure 4G). 

The SGs adjacent to the APM were able to fuse with it under the influence of an external stimulus. A hole appeared and cationized ferritin or gold entered it. At the moment of opening a large pore, the goblet cell can capture a portion of fluid from the intestinal lumen. It enters autophagosomes and lysosomes. After the release of mucus, the membrane of the first SG undergoes endocytosis, and the membranes can enter the GC distensions, as shown by Nikonova et al. [65]. These membranes can then be secreted via the BLPM. There are always two or three dendritic cells at the base of the goblet cell; their processes pass through the VM and come into contact with lymphocytes, and not with all B-lymphocytes.

The inhibition of clathrin-mediated endocytosis leads to the accumulation of SGs. The exocytosis of mucin from SGs is regulated by intracellular Ca^2+^ levels, and Ca^2+^ mobilizing agents, such as acetylcholine and histamine, are powerful inducers of intestinal mucus secretion. Acetylcholine is by far the most studied agent for mucus secretion, and it has been shown to induce mucus secretion in both the small and large intestines of mice, rats, and rabbits, as well as in the human colon. Stimulation with acetylcholine or other cholinergic agonists, such as carbachol, leads to a rapid short-term increase in the rate of mucus secretion, which returns to its original levels within 30 min. The population of the small intestine can also acquire lumen antigens and present them to CD103+ dendritic cells in its own plate. Mucus renewal occurs much faster than epithelial cell renewal and is most likely closely related to the advancement of lumen material [33].

Goblet cells in the small intestine deliver lumen antigens to a cluster of differentiating dendritic cells through channels associated with goblet cells [27]. It is argued [26,27] that, in response to acetylcholine, goblet cells undergo a considerable endocytic process that transports a load of liquid phase through the cell for delivery to dendritic cells (which are in close contact with goblet cells, and then deliver antigens to mononuclear phagocytes of their own plates) (Figure 5A–E). 

Indeed, under the action of acetylcholine, the passage of an antigen associated with goblet cells through an endocytic event was observed, “which effectively delivers cargo in the liquid phase not only to lysosomes, MVBs and TGN, but also to the transcytosis pathway, which allows the lumen substances to be captured by underlying cells” (p. 17 in [26]). It includes the retrograde delivery of liquid-phase cargo to the trans-Golgi network and the secretion of this cargo, containing multivesicular bodies and lysosomes, in the space between enterocytes (see Figure 2B,F,G, which are represented by Gustafsson et al. [26]). This delivery is not required to maintain the mucosal barrier. The inhibition of clathrin-mediated endocytosis, as well as defects in the proteins associated with autophagy, induces the aggregation of the mucin granules of goblet cells [79]. Its blockade does not suppress mucus secretion [26]. The visualization of mucus production and goblet cells in a monolayer obtained from the large intestine of a dog revealed a fenestrated membrane extending deep into the cell and microvilli [80].

However, our analysis of images presented in the literature and on the Internet showed that there was no apical endocytosis in goblet cells. Additionally, we did not find clathrin-coated buds on APM goblet cells. Moreover, our study of the images presented by Gustafsson et al. [26] showed that, on all of them, these secreted structures looked similar to autophagosomes of multilayer bodies involved in the early formation of autophagosomes [81,82].

## 7. Models of Intracellular Transport and Regulation Secretion

The GC scheme in goblet cells is shown in Figure 5. The RS provides a convincing example of the difficulties associated with the vesicular model (VM; described in detail in [83,84,85,86,87,88]; also, see schemes in Figure 6I–III). Significant difficulties arise due to the phenomenon of non-parallel transfer [68,89]. RS proteins are diffusive because the rate at which molecules move is determined by their own concentration and mobility, and not by an external factor (for example, vesicles), and they are based on equilibrium because they are reversible and, as such, allow for transport in both directions through a given membrane, and not just in one, as in the case of vector vesicular mechanisms. They also differ from vesicle-based mechanisms in that the molecules (1) usually move individually rather than in groups, (2) reach the membrane through the solvent phase (e.g., cytosol) rather than through vesicles, and (3) pass through intact membranes rather than through disturbances caused by cutting or embedding the membrane (see Figure 2 presented by Rothman [68]). Thus, VM cannot be applied, since the concentration of cargo in them is less than in cisternae, and the VM cannot explain the parallel secretion. To date, no study has demonstrated that the concentration of RS proteins in COPI is significantly higher than in Golgi cisternae. The VM is also not suitable because there are only very few COPI vesicles.

On the other hand, the enrichment of RS proteins on the trans-side of the GC is an unresolved issue for the cisternal maturation progression model [2,83,84,85,86,87,88]. Thus, the CMPM is not suitable because there is a concentration of mucus on the trans-side of the GC. This phenomenon should be avoided within the framework of the CMPM [44,83].

The secretory cycle hypothesis was proposed by S. Rothman [68], which poses that, after synthesis in the ER, secreted proteins diffuse into the cytosol, and move from it into secretory granules for storage, and are then diffused from the cytosol through the APM into the lumen of the glands or intestines. When the transport through the cytosol is refuted, the diffusion model can be very useful for explaining the transport of proteins that accumulate in secretory granules that are released under the action of an external command to the cell. Large pleomorphic post-Golgi carriers are formed as a result of the maturation of the trans-Golgi compartment [84,90].

S. Rothman [68] wrote: “… the protein synthesised in the ER should … exit by budding the membrane, because there is no other option.” However, such an option exists, at present, in the form of the “kiss and run” model (KARM). It is based on the assumption that several separate compartments coexist in the secretory pathway and that membrane fusion events along with successive divisions can be used to form the maturation of compartments. In most cases, regulated secretory proteins must be converted from a condensed to soluble form after the fusion of SGs with plasmalemma. To make such a system work: (1) compartments with different ionic environments should be narrowly connected to limit mixing and (2) the concentration of various RS proteins along these interconnected, but different, compartment domains should create a diffusion gradient along tubular compounds [83,85,86]. It seems that the KARM can explain the biogenesis and maturation of immature SGs [2,91,92]. However, the number of rims capable of serving as a mechanism for the progression of cargo domains is low, and there is no stretching within the edges of the first and second Golgi cisternae. Therefore, we assumed that the Golgi stacks formed some kind of a spiral. One of the possibilities could be the diffusion of cargo along the spiral GC proposed by Tanaka et al. [93].

Due to the organization of Golgi stacks in the form of a spiral, RS proteins acquire more and more polysaccharide chains during their diffusion along the spiral Golgi stacks, and, finally, acquire the ability to form dynamic aggregates. This leads to the formation of a distension at the rim of the third cisterna. Diffusion induces the intake of an increasing amount of mucin, and in the central part of the Golgi cisternae, cisternal distensions begin to form. Finally, the entire lumens of the fifth and sixth Golgi cisternae are filled with mucin aggregates. It seems that such cisternae are transformed into immature SGs. Following secretion, the membranes of empty SGs can undergo considerable clathrin-independent endocytosis and are then transformed into autophagy-dependent multilayer organelles (MLOs), which are delivered to the GC and can be secreted into the space between goblet cells and enterocytes, where dendritic cells await them. On the other hand, MLO membranes can be delivered to the lysosomes. There, lipids can be cleaved and transferred to the ER through ER–lysosome contact sites. These fatty acids can be used for the formation of Golgi membranes (Figure 5E). We hypothesized that, in goblet cells, the membranes are almost not synthesized because there no smooth ERs exist. As such, the membranes of already-formed SGs can be used. As this cell moves to the tip of the intestinal villi, its volume and the surface area of the membranes decrease, that is, following maturation, the membranes are used until it reaches the very tip of the villus and peels off together, with the surrounding enterocytes containing brush borders. Due to the above-mentioned events, goblet cells can perform the transcytosis of IgA from their own plate into the lumen of the intestine and respiratory tract. In any case, the observation of MLOs inside the cisternal distensions indicates that these MLOs, together with the protein attached to or inserted into their membranes, can be secreted into the space between epithelial cells.

## 8. Conclusions and Future Perspectives

Here, we highlighted the issue related to RS and intracellular transport in goblet cells. Following the synthesis, transportation, and accumulation of the secretion, SGs wait for the signal to be released. After the APM ruptures, due to fusion between the APM and the membrane of the secretory granule tightly pressed against it, calcium enters the cell, since the secretory granules are apparently not sealed; then, a considerable fusion of some granules with each other occurs. Some of these phagocytes move through the lymphatic capillaries to the lymph nodes, where they induce adaptive immune responses that promote the induction of tolerance to the contents of the lumen [27].

The membrane of the first fused SG becomes part of the APM, and then undergoes endocytosis and turns into an autophagosome. Substances also enter the granules from the intestine. COPI-dependent vesicles are accumulated between the cisternae and SGs during the maturation of goblet cells, and then these vesicles are used as a source of SNAREs for the fusion of SGs. Most likely, SNAREs are not very important for the fusion of SGs because the mechanism of SNARE-recycling remains unclear. The modified “kiss and run” model can be equally applicable to earlier intracellular compartments along the anterograde transport pathway, and can be closely related to the Golgi structure and sorting mechanisms for SG biogenesis.

It seems that goblet cells can also perform the transcytosis of IgA from the interstitial space into the lumen of the intestine, respiratory tract, or other ducts. Therefore, the labeling of IgA, COPII, SNAP29, and membrin/GS27 can be very useful. It is not completely clear whether GC stacks form a complete spiral. It is desirable to perform a complete reconstruction of the GC in goblet cells, based on the three-dimensional electron microscopy with a high resolution [94].

After analyzing the literature and additional analysis of our data, many questions arise. If we assume that Golgi transport is performed by COPII-dependent vesicles, then why are there no COPII-coated buds on the ER? The connections with the emergency department do not seem to be evident. If transportation is conducted on the basis of the maturation scheme, it should be tested whether there are resident Golgi proteins in COPI vesicles. It is unclear why these bubbles accumulate between cisternae and secretory granules. Are they dependent on COPI or clathrin? It is not clear whether Golgi enzymes and membrin/GS27 are present in the membranes of COPI vesicles in the goblet and pancreatic acinar epithelial cells.

Where are the proton pumps located in the membrane obtained from the emergency department from the very beginning? The matrix between the Golgi cisternae is thinned and then disappears. Where do the membranes come from and how are they processed? Where do the membranes of secretory granules go after mucus secretion? Can it be endocytosis or autophagy, as in the acinar cells of the pancreas? Perhaps, the pellets are being restored and fused with the Golgi cisternae. How does the fission of the filled secretory granule (the previous Golgi cisterna) occur? Where are fatty acids synthesized? Are they reused? Is apical endocytosis observed in goblet cells? Are autophagosomes and multilamellar organelles secreted into the space between epithelial cells? What is the fate of goblet cells after the secretion of all RS proteins, if such a process exists? How are autophagosomes delivered via BLPM to the surrounding goblet cell by dendritic cells? Are goblet cells one-use items? What are the mechanisms and purpose of this phenomenon? What is the mechanism responsible for the coalescence of cisternal extensions and their transformation into secretory granules? What is the role of the ER localized around the SG aggregate? Where do monosaccharides necessary for glycosylation come from? Are hydrogen bonds responsible for the concentration of mucus due to acidic residues of sialic acid? Do membrane residues formed after the secretion and endocytosis of SGs enter autophagosomes? Are there secretion, accumulation, and recovery phases? Where do their membranes come from? How is APM restored? Is the secret revealed all at once or in parts. If in parts, then how is APM is restored; the same question arises if the whole secret is revealed at once. Does the GC stack really form a spiral? 

## Figures and Tables

**Figure 1 ijms-24-09560-f001:**
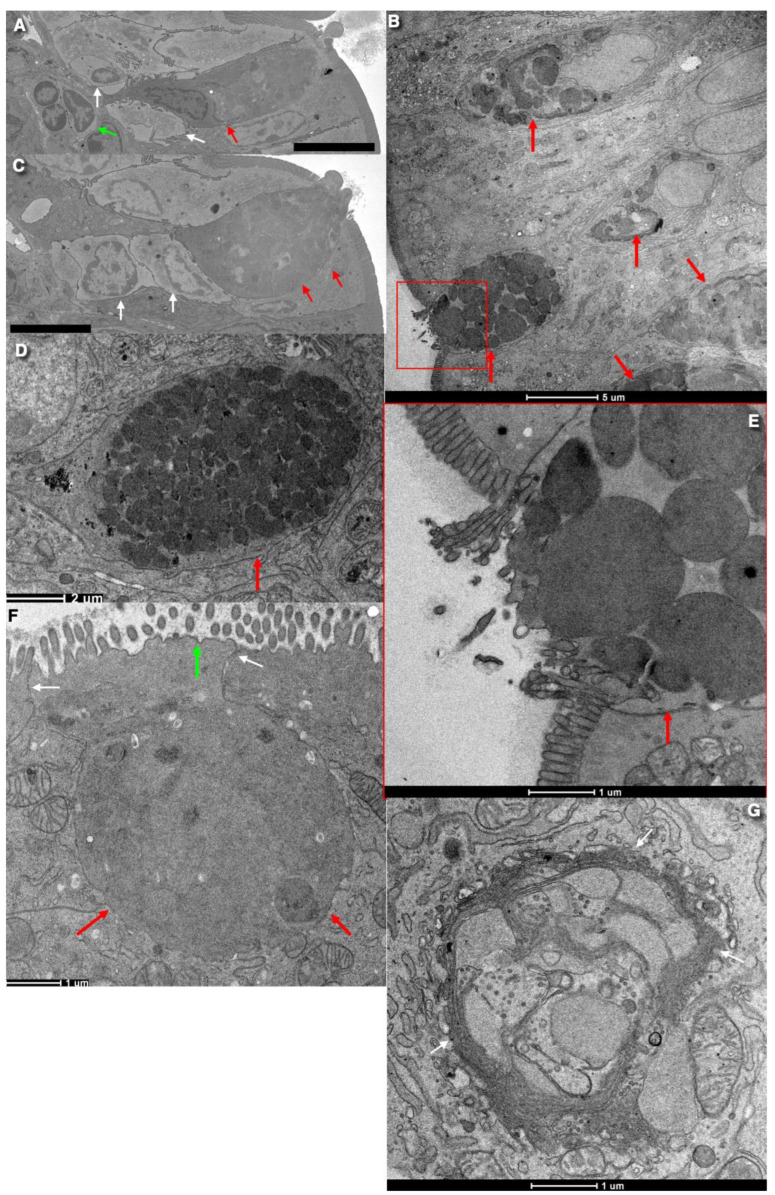
Ultrastructure of goblet cells. (**A**–**D**) Goblet cells in jejunum in different stages of their life cycle. (**A**,**C**) Serial images obtained with the help of 3VIEW show the goblet (red arrows) and dendritic (white and green arrows) cells near the basis of the goblet cell. (**E**) The initial phase of mucus secretion. Enlarged area situated within the red box in (**B**). Goblet cells before secretion. (**F**) Goblet cells (red arrows) before secretion. Green arrow shows apical microvilli. White arrows show right junctions. Goblet cells (red arrows) in different stages of their life cycle taken from the Figure 3A presented by Briata et al., 2023 [25]. (**G**) Cross-section of the Golgi area just above the bottom of the “Golgi cap”. Scale bars: 7 (**A**,**C**); 5 (**B**); 1 (**E**,**F**); 2 µm (**D**).

**Figure 2 ijms-24-09560-f002:**
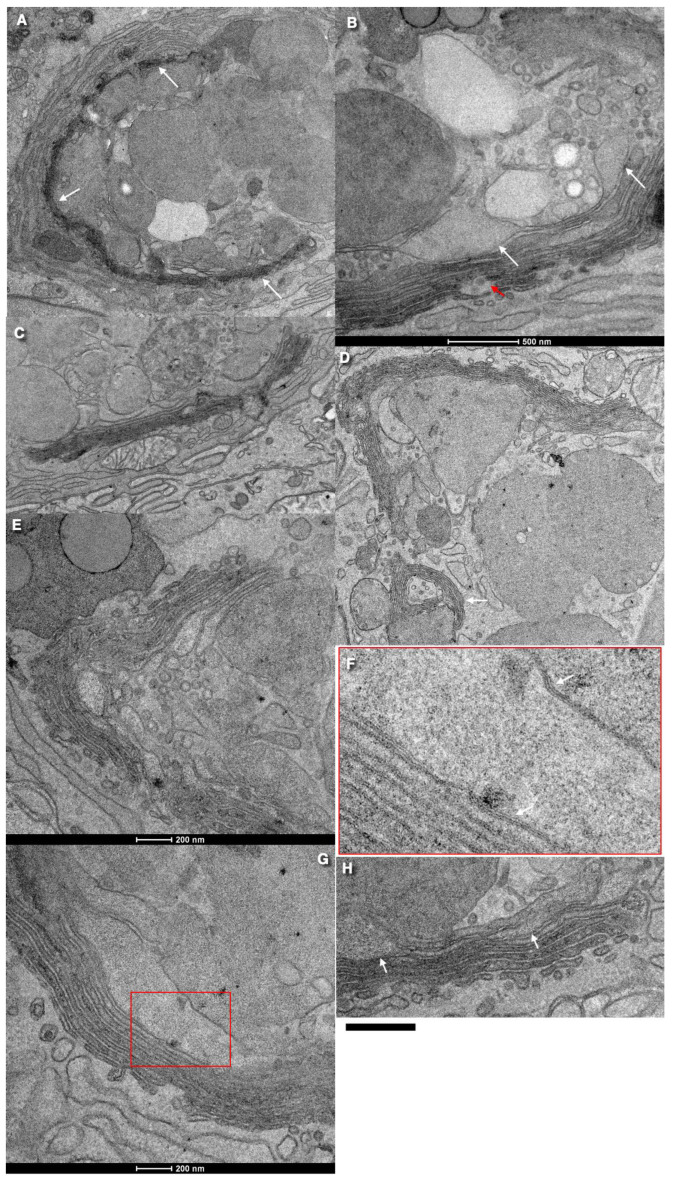
Structure of the GC in the goblet cells of the colon (**A**–**E**,**G**,**H**). (**F**) High magnification of the images inside the red box in (**G**). White arrows show pentalaminal structure of membrane contact between SGs. Fragment of the image in Figure 3A presented by Briata et al. [25]. Scale bars (nm): 200 (**E**,**G**,**H**); 500 (**C**); 1000 (**A**,**C**,**D**).

**Figure 3 ijms-24-09560-f003:**
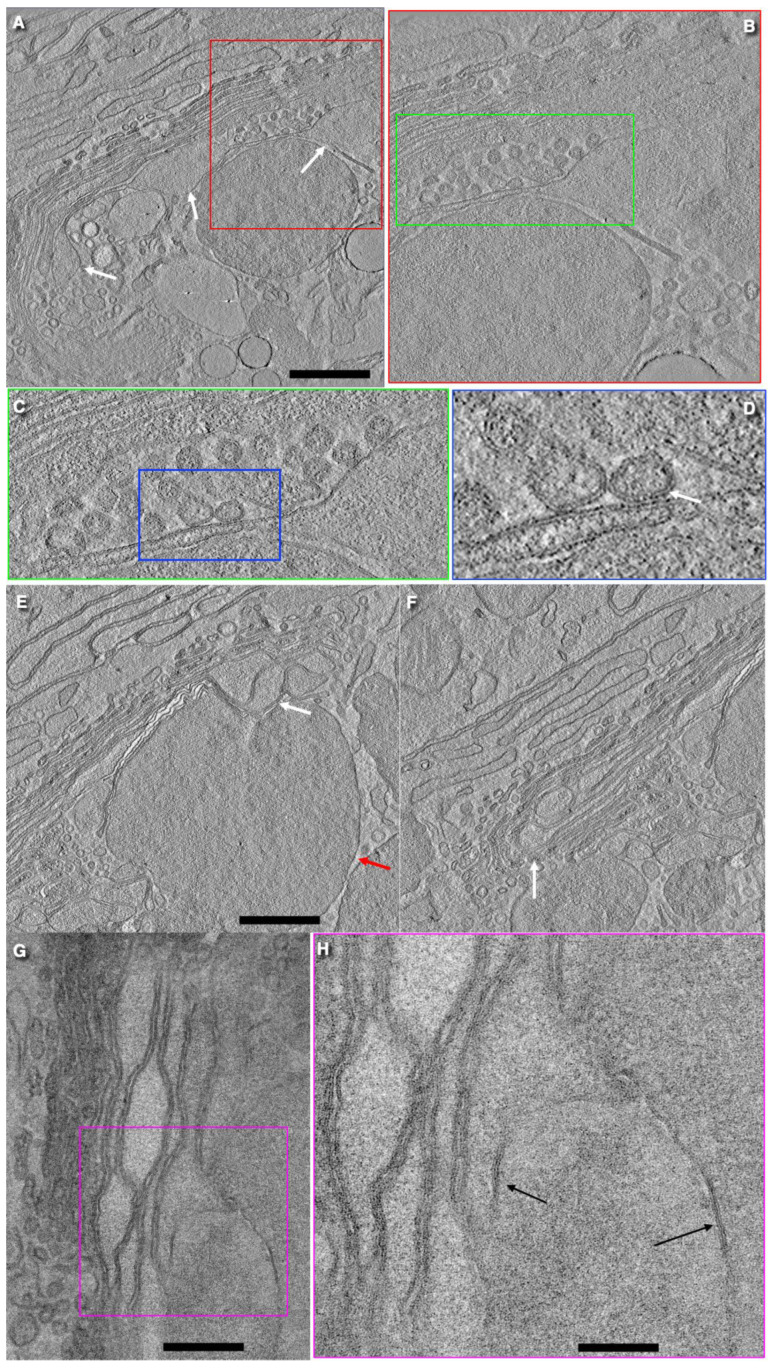
Fine organization of the GC of goblet cells. (**A**) Medial Golgi cisterna with 3 distensions (white arrows). (**A**–**D**) Consecutive images with increasing magnification show clusters of COPI-dependent vesicles. Some of these vesicles (**D**) are tightly attached to the membrane of medial Golgi cisternae. The image of Golgi vesicular aggregates shown in (**A**–**D**) is taken from Figure 3Ae presented by Briata et al. [25]. These images represent a serial section of Figure 3Ae, but are obtained with the help of electron microscopic tomography. (**E**) Cluster of Golgi cisternal distensions (white arrow) near the cisternal rims. Red arrow shows SGs. (**F**) Bending of the Golgi stack (white arrow). (**G**,**H**) Close contacts between SGs (pentalaminal contact). Scale bars: 710 (**A**,**E**–**G**); 360 (**B**); 190 (**C**); 90 nm (**D**,**H**).

**Figure 4 ijms-24-09560-f004:**
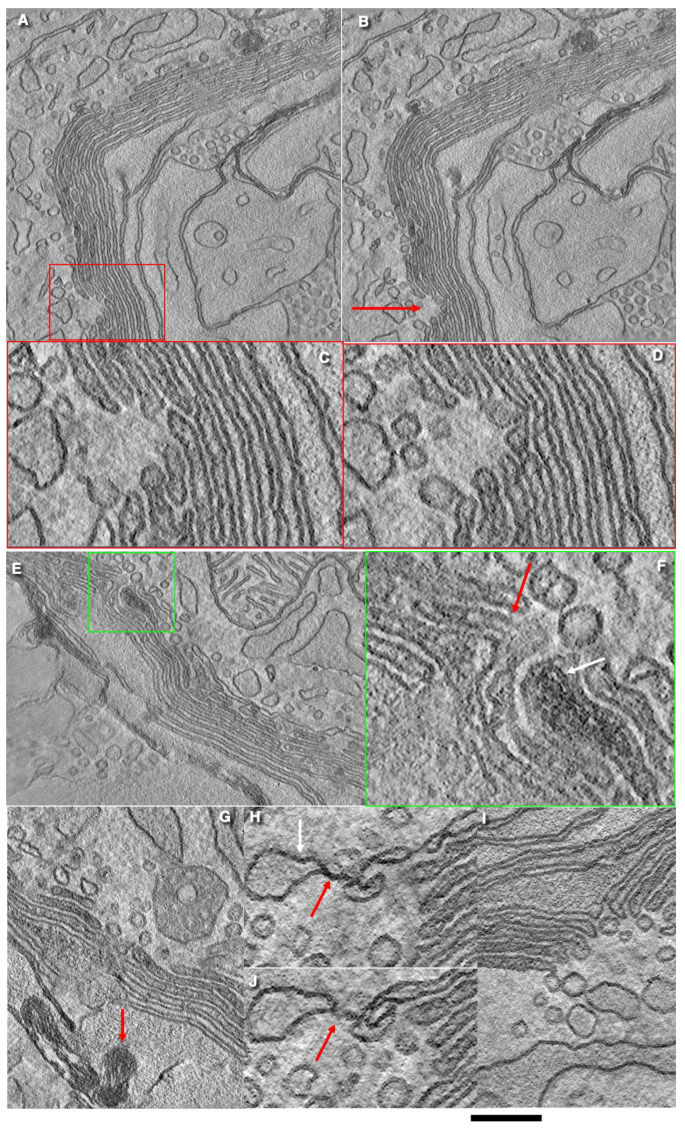
Tomography slices of the GC in goblet cells show a well (red arrow in (**B**)). (**A**,**B**) Serial tomoslices show the varicosities presenting connections between the ER and Golgi cisternae. (**C**,**D**) Enlarged areas in serial images shown in the red box in (A and corresponding area in (**B**). (**E**,**F**) Formation of spiral by Golgi cisternae within the well. (**G**) Autophagy elements (red arrow) between SGs. (**H**–**J**) Thin connections between the ER and Golgi cisterna (red arrows). White arrow indicates the ER. Scale bars (nm): 400 (**A**,**B**); 130 (**C**,**D**); 350 (**E**,**G**,**I**); 100 (**F**,**H**,**J**).

**Figure 5 ijms-24-09560-f005:**
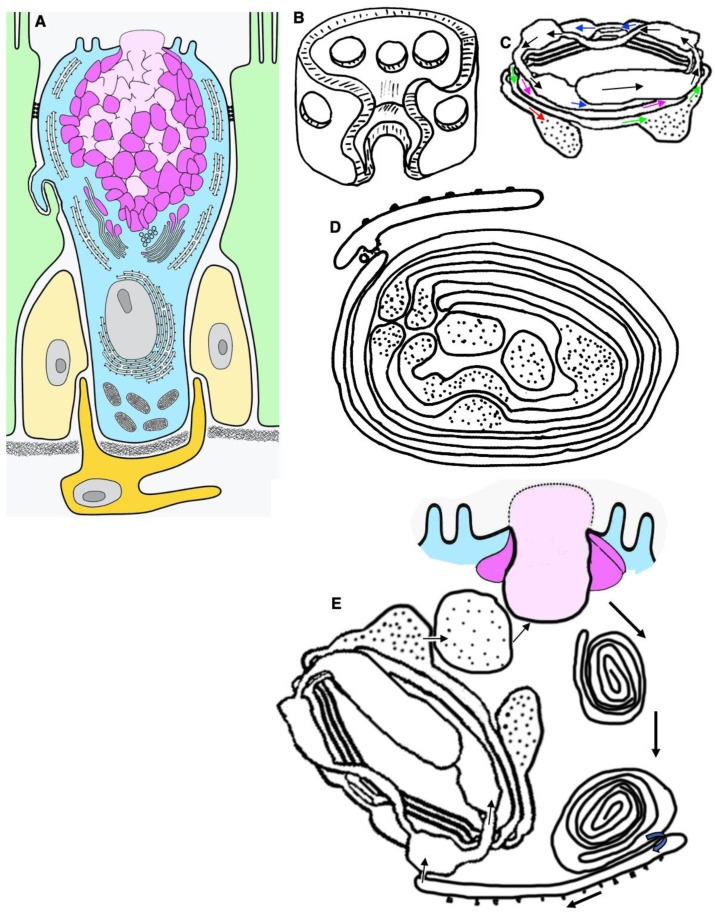
Scheme of the secretory pathway of a goblet cell. (**A**) The GC is below the SG aggregates (magenta). Nucleus is colored in aqua. (**B**) Scheme of the “Golgi cap”. (**C**,**D**) Schemes of the Golgi spiral. (**E**) Possible membrane turnover during the secretion cycle. It includes the secretion of mucus, non-clathrin-dependent endocytosis of the membrane of SGs, its transformation into autosomes or phagophores, the transport of fatty acids from phagophores (or lysosome-containing phagophores) into the ER membrane, and then the delivery of these membranes to the GC.

**Figure 6 ijms-24-09560-f006:**
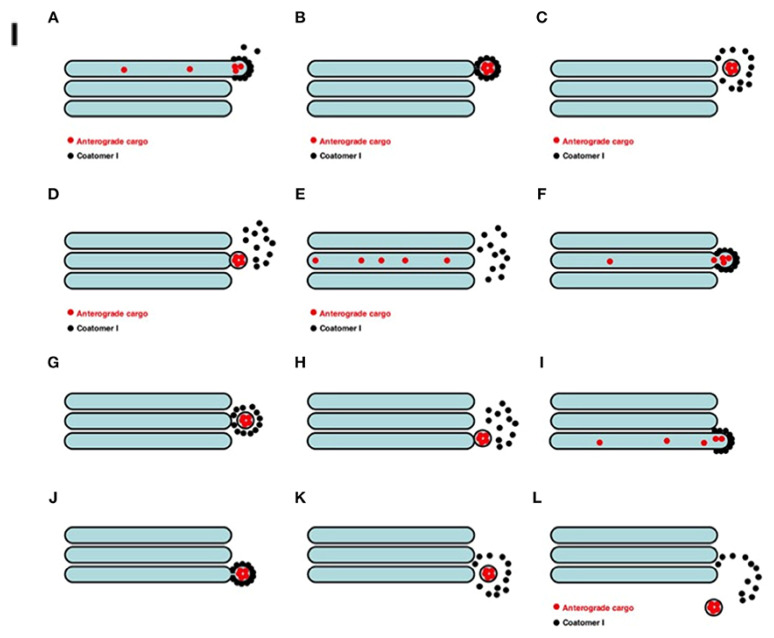
Schemes of models of intra-Golgi transport (**I**–**III**) obtained from Figures 5, 7, and 8 (correspondingly), presented by Mironov and Beznoussenko [83]. (**I**) Scheme of the vesicular model for intra-Golgi transport (see also https://yadi.sk/i/M1ykjxKabmjs5 accessed on 21 May 2023). (A) COPI (black dots) forms a coat on a membrane bud. Cargo (red dots) is concentrated inside the COPI-coated bud. (B,C) This bud undergoes detachment (B) and uncoating (C). (D) The vesicle fuses with the distal Golgi compartment. (E) Distribution of the cargo within the next Golgi cisterna. (F–H and I–L) Repetition of the first stage. Finally, the vesicle can exit from the Golgi (L). (**II**). Scheme of intra-Golgi transport according to the cisterna maturation-progression model. The main idea of this model is that, during intra-Golgi transport, the amount of cargo inside the cisterna during its progression does not change, and that COPI vesicles (COPI: black dots; Golgi-resident proteins: colored dots) should be concentrated in COPI vesicles. (A) Formation of ER-to-Golgi carriers (top). (B) Delivery of ER-to-Golgi carriers to the Golgi complex. (C) Fusion of ER-to-Golgi carriers and formation of the new *cis-*Golgi cisterna. (D) Formation of COPI (black dots)-coated buds on Golgi cisternae. (E) Detachment of buds and their uncoating. (F) COPI-dependent vesicles fuse with proximal Golgi cisternae. (G–I) A new round of *cis-*cisterna formation, COPI-dependent budding, formation of vesicles, and their uncoating and fusion. (I) Departure of the most trans-cisterna in the form of post-Golgi carriers. (J–N) Additional rounds of similar events. (O) After the stepwise departure of post-Golgi carriers, the *cis-*Golgi cisterna formed after the re-initiation of IGT becomes the *trans-*cisterna. (**III**) Scheme of intra-Golgi transport according to the kiss-and-run model. (A–H) Its main principle is the following: initially there is SNARE-dependent fusion with the distal Golgi cisterna, and then fission along the line of the pore row occurs. (I–Q) Scheme showing only two cisternae in three-dimensional form. (I) Formation of the cargo domain (red arrow) separated by the row of pores. (J) Fusion of the domain with the distal cisterna and enlargement of pores. (K) Break down of two tubules surrounding the pores. (L) Break down of the last tubule connecting the domain to the proximal cisterna. (M–Q) Stages of pore formation when the membrane bud grows and then fuses backward to the same cisterna. During the transport wave, there is the consumption of the cisternal pores.

## Data Availability

Not applicble.

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
