# Peer review of "The Regulated Secretion and Models of Intracellular Transport: The Goblet Cell as an Example"

_ijms, 2023, doi:10.3390/ijms24119560_

Round 1

Reviewer 1 Report

A review article on modes of secretion and aspects of intracellular transport in Goblet cells is presented. In general, the subject is interesting and the manuscript summarizes important information that may prove useful for people interested in this field. Below are some comments for consideration.

In order to increase the readers’ ability to follow the presentation, it would be very helpful and important to create and include graphical schemes, which would accompany the relevant chapters and highlight the referenced pathways and models, perhaps also including some key features at the molecular level. This would be particularly helpful for the “intracellular transport models”, or the “morphology of the RS pathways”, or aspects of the “kiss-and-run” vs other hypotheses. In general, the existing figures and the overall graphical presentation should be improved.

Line 27-28: It is not clear to which issue of IJMS the author is refereeing to. Perhaps this sentence should be omitted.

Line 48-49: More information is necessary if the author wants to cite this article. Is it the Briata et al. (2023) which does not yet exist in the databases?

Author Response

Reviewer 1

A review article on modes of secretion and aspects of intracellular transport in Goblet cells is presented. In general, the subject is interesting and the manuscript summarizes important information that may prove useful for people interested in this field. Below are some comments for consideration.

  1. In order to increase the readers’ ability to follow the presentation, it would be very helpful and important to create and include graphical schemes, which would accompany the relevant chapters and highlight the referenced pathways and models, perhaps also including some key features at the molecular level. This would be particularly helpful for the “intracellular transport models”, or the “morphology of the RS pathways”, or aspects of the “kiss-and-run” vs other hypotheses.

Reply: We included several schemes of the models. We eliminated the part on pancreatic acinar cells. This made the text more focused. Also we include some sentences on molecular mechanisms.

  1. In general, the existing figures and the overall graphical presentation should be improved.

Reply: We improved images and added several new images. Quality of our images is high because in each image we resolve three layers of any membrane. It is well-visible in our images 3 and 4. In works published recently such high quality of EM images could not be found.

  1. Line 27-28: It is not clear to which issue of IJMS the author is refereeing to. Perhaps this sentence should be omitted.

Reply: the sentence is eliminated.

  1. Line 48-49: More information is necessary if the author wants to cite this article. Is it the Briata et al. (2023) which does not yet exist in the databases?

Reply: finally this paper is published and included into the reference list.

Reviewer 2 Report

In this review, the author aims to summarize the regulated secretion in goblet cells,  which is not well explored in the past. Overall, the manuscript will be interesting to cell biologists. However, the manuscript is not well organized, which makes it not friendly for the readers. Moreover, there are too many obvious spelling and grammar mistakes that must be corrected, and improvements in writing should be done before acceptance to publish in the International Journal of Molecular Sciences.

Specific points:

1.    The high resolution of Golgi apparatus in Goblet cells from both EM and IF will be good to include.

2.    The mechanism of intracellular transport (like mucins) in Goblet cells is still not very clearly shown. What is the similarity and difference between the RS pathway of Goblet cells and other secretory cell types (like pancreatic b cells or acinar cells)?

3.    Are there any major players (proteins) that are key to regulating the switching on/off RS?

4.    In lines 38-40, it is redundant for the two sentences about the RS in endocrine.

5.    In line 72, no body demonstrated an isolated ministack, which is overstated. The recent paper (PMCID: PMC100636060) showed single Golgi stacks in mice pancreatic cells, even though not by 3D but 2D images also help.

6.    In line 74, stack should be stacks.

7.    The paragraph from line 78 to line 84 seems not related to the last and next paragraph.

8.    In the paragraph from line 83 to line 89, why are there two maturation steps of SGs? The last sentence is not complete. “Life cycle” seems not appropriate to describe RS.

9.    In line 107 and line 256, that should be added between “than in”; in line 143, induce should be induces; in line 194, there should be deleted; in line 202, as as, one as should be deleted; in line 216, here should be deleted; in line 264, not should be no, are should be is; in line 277, is low should be deleted; in line 288, allow should be allows; in line 327, Hthere, should be corrected; in line 342, two acinar should be left one; in line 388, It should be it; the line 410 is confusing, open secretory granules, other SG? In line 484, EP is ER?

10. Line 130 is redundant with line 175.

11. In line 197, Usually the GC is composed of seven cisternae, which is not precise. In most literature, there are 4-6 cisternae in each Golgi stack.

Author Response

Reviewer 2

In this review, the author aims to summarize the regulated secretion in goblet cells,  which is not well explored in the past. Overall, the manuscript will be interesting to cell biologists.

  1. However, the manuscript is not well organized, which makes it not friendly for the readers.

Reply: we reorganized the text, made it more focused and eliminated several its parts.

  1. Moreover, there are too many obvious spelling and grammar mistakes that must be corrected, and improvements in writing should be done before acceptance to publish in the International Journal of Molecular Sciences.

Reply: we eliminated most mistakes. Moreover during editing many previous words were replaced and not it is very difficult to find our previous mistakes.

Specific points:

  1. The high resolution of Golgi apparatus in Goblet cells from both EM and IF will be good to include.

Reply: we included new images according the reviewer request. Our morphological analysis used for publication of our paper by Sesorova et al., (2020) and Briata et al (2023) did not include IF analysis of the GC with high resolution . Therefore we only indicated papers where such kind of images (although rather r limited) the reader could find.

  1. The mechanism of intracellular transport (like mucins) in Goblet cells is still not very clearly shown. What is the similarity and difference between the RS pathway of Goblet cells and other secretory cell types (like pancreatic ß cells or acinar cells)?

Reply: we included these explanations and schemes of the KARM and other models of intra-Golgi transport related to RS.

  1. Are there any major players (proteins) that are key to regulating the switching on/off RS?

Reply: we added some molecular mechanisms into the text.

  1. In lines 38-40, it is redundant for the two sentences about the RS in endocrine.

Reply: we corrected the text.

  1. In line 72, nobody demonstrated an isolated ministack, which is overstated. The recent paper (PMCID: PMC100636060) showed single Golgi stacks in mice pancreatic cells, even though not by 3D but 2D images also help.

Reply: indeed, isolated Golgi stacks could be seen in pancreatic β cells (Fig. 1B presented by Muller et al., 2020). However, these cells are subjected to RS through BLPM not through APM as in goblet and exocrine pancreatic cells which secret RS protein through the APM. In acinar epithelial cells these stacks are absent (Fig. 6 presented by Koga et al., 2016).  Also we and Gustafsson and Johansson (2022) could not find such stacks in goblet cells. We have included only limited information of molecular machines because only few people have seriously dealt with goblet cells and many questions about them are solved on the basis of the "by analogy" method.

  1. In line 74, stack should be stacks.

Reply: We corrected this.

  1. The paragraph from line 78 to line 84 seems not related to the last and next paragraph.

Reply: we changed the place for this paragraph.

  1. In the paragraph from line 83 to line 89, why are there two maturation steps of SGs? The last sentence is not complete. “Life cycle” seems not appropriate to describe RS.

Reply: we corrected these parts of the text.

  1. In line 107 and line 256, that should be added between “than in”; in line 143, induce should be induces; in line 194, there should be deleted; in line 202, as as, one as should be deleted; in line 216, here should be deleted; in line 264, not should be no, are should be is; in line 277, is low should be deleted; in line 288, allow should be allows; in line 327, "Hthere", should be corrected; in line 342, two acinar should be left one; in line 388, It should be it; the line 410 is confusing, open secretory granules, other SG? In line 484, EP is ER?

Reply: we corrected these parts of the text. Several parts of the text which sometimes include our previous mistakes were elim8nated. Moreover during editing many previous words were replaced and not it is very difficult to find our previous mistakes. The text, which contains corrected mistakes, before its total reorganization is shown in the end of the PDF variant.

  1. Line 130 is redundant with line 175.

Reply: we eliminated one of these lines.

  1. In line 197, Usually the GC is composed of seven cisternae, which is not precise. In most literature, there are 4-6 cisternae in each Golgi stack.

Reply: indeed, in most cells in culture the GC is composed of 4-6 medial cisternae. However, nobody is considered that the cis-most and the trans-most cisternae also could be counted (Mironov and Beznoussenko 2008; Mironov et al., 2017). In our text, which we corrected, and indicated that this number is related to the GC in goblet cells.

Round 2

Reviewer 2 Report

In the revised version, the manuscript is greatly improved. Spelling and grammar mistakes that must be corrected, and improvements in writing should be done before acceptance to publish in the International Journal of Molecular Sciences.

Specific points:

1.    For example, in lines 58, 76, 106, 172, and 182, one space should be deleted. In line 135, “H-linked glycosylation” should be N-linked…

Author Response

Our Reply

Reviewer 3.

In the revised version, the manuscript is greatly improved. Spelling and grammar mistakes that must be corrected, and improvements in writing should be done before acceptance to publish in the International Journal of Molecular Sciences.

Rely: We checked our English again and corrected all mistakes.

Specific points:

  1. For example, in lines 58, 76, 106, 172, and 182, one space should be deleted. In line 135, “H-linked glycosylation” should be N-linked…

Reply: We corrected these mistakes.